# P300 to Low and High Frequency Stimuli Are Not Influenced by Intensity in Adults with Normal Hearing

**DOI:** 10.3390/brainsci15020209

**Published:** 2025-02-18

**Authors:** Giulia Cartocci, Garrett Cardon, Julia Campbell, Bianca Maria Serena Inguscio, Dario Rossi, Fabio Babiloni, Anu Sharma

**Affiliations:** 1Department of Molecular Medicine, Sapienza University of Rome, Piazzale Aldo Moro 5, 00185 Rome, Italy; dario.rossi@uniroma1.it; 2BrainSigns Ltd., Via Tirso, 14, 00198 Rome, Italy; biancams.inguscio@uniroma1.it (B.M.S.I.); fabio.babiloni@uniroma1.it (F.B.); 3Communication Disorders Department, Brigham Young University, Provo, UT 84602, USA; 4Department of Speech, Language, and Hearing Sciences, University of Texas, Austin, TX 78712, USA; julia.campbell@austin.utexas.edu; 5Department of Computer, Control, and Management Engineering, Sapienza University of Rome, Piazzale Aldo Moro 5, 00185 Rome, Italy; 6Department of Physiology and Pharmacology “Vittorio Erspamer”, Sapienza University of Rome, Piazzale Aldo Moro 5, 00185 Rome, Italy; 7Department of Speech Language and Hearing Sciences, University of Colorado, Boulder, CO 80309, USA; anu.sharma@colorado.edu

**Keywords:** EEG, hearing loss, P300, cortical auditory evoked potentials, oddball, high frequency, low frequency, warning cues, obligatory responses, auditory cognitive processing

## Abstract

**Background/Objectives**: Since high frequencies are susceptible to disruption in various types of hearing loss, a symptom which is common in people with tinnitus, the aim of the study was to investigate EEG cortical auditory evoked and P300 responses to both a high- and low frequency-centered oddball paradigm to begin to establish the most suitable cognitive physiologic testing conditions for those with both unimpaired hearing and those with hearing impairments. **Methods**: Cortical auditory evoked potential (CAEP) P1, N1, P2 and P300 (subtraction wave) peaks were identified in response to high- (standard: 6000 Hz, deviant: 8000 Hz) and low frequency (Standard: 375 Hz, Deviant: 500 Hz) oddball paradigms. Each paradigm was presented at various intensity levels. Latencies and amplitudes were then computed for each condition to assess the effects of frequency and intensity. **Results**: Stimulus intensity had no effect on either the high- or low frequency paradigms of P300 characteristics. In contrast, for the low frequency paradigm, intensity influenced the N1 latency and P2 amplitude, while for the high frequency paradigm intensity influenced P1 and P2 latency and P2 amplitude. **Conclusions**: Obligatory CAEP components responded more readily to stimulus frequency and intensity changes, and one possible consideration is that higher frequencies could play a role in the response characteristics exhibited by N1 (except for N1 amplitude) and P2, given their involvement in attentional processes linked to the detection of warning cues. P300 latency and amplitude were not influenced by such factors. These findings support the hypothesis that disentangling the cognitive from the more sensory-based response is possible, even in those with hearing loss, provided that the patient’s hearing loss is considered when determining the presentation level. While the present study was performed in participants with unimpaired hearing, these data set up future studies investigating the effectiveness of using similar methods in hearing-impaired persons.

## 1. Introduction

P300 event-related potential (ERP) is a commonly employed EEG-based measure of cognitive function and attention. This ERP component is represented by a large, positive waveform peak that typically occurs around 300 ms after the onset of a rare, task-relevant stimulus. This ERP typically has a centro-parietal scalp distribution that is maximal over midline scalp sites [1]. Traditionally, P300 is supposed to be relatively immune to the effects of stimulus factors because of its putative endogenous rather than exogenous origin, whilst stimulus parameters affect the amplitude and latency of exogenous, obligatory potentials (e.g., N1, P2, N2) [2]. According to the context updating theory for P300, following the initial sensory processing of the stimulus, an attention-driven comparison process occurs to assess the representation of the previous event in working memory. Then, whether no change is detected in terms of the stimulus attributes, so only sensory evoked potentials or cortical evoked potentials (CAEP) are recorded (N1, P2, N2); on the contrary, when a new stimulus is detected, attentional processes “update” the stimulus representation corresponding to P300 [3]. Moreover, it appears that P300 amplitude is more strictly related to attentional resource allocation, while P300 latency is more closely related to the detection and evaluation of a target stimulus [3].

Concerning obligatory cortical activity occurring in response to auditory stimuli, the P1 is an early positive component originating from the auditory cortex. The P1 shares its source with the slightly later obligatory auditory evoked potentials N1 and P2 (e.g., [4,5,6]). These three potentials form the P1-N1-P2 complex. N1, one of the most studied CAEPs in adults, reflects auditory detection and discrimination [7]. Additionally, P2 can be altered by various participant- and environment-related characteristics in adults without hearing impairments, such as age, sleep, and respiratory stimuli [8]. Recently, some researchers have provided evidence that the N1 and P2 are generated independently [8]. Germane to the current report, despite N1-P2 appearing to be influenced by arousal level, it is not clear whether these potentials are associated with cognitive processing [9].

Evoked potentials are commonly used to test those with hearing loss. However, much of the focus of such physiological testing rests on subcortical measures (e.g., auditory brainstem response—ABR). Though they are used less frequently in clinical settings, CAEPs have shown great utility in evaluating the developmental status of the auditory cortex [10,11], speech detection and perception (e.g., [12,13]), estimating auditory threshold (e.g., [9,14]), determining the effectiveness of both hearing aid and cochlear implant use (e.g., [14,15,16,17,18]), among other functions. Fortunately, this pattern of less frequent use of CAEPs may be improving in more recent years (e.g., [19]). Along with such changes, it is reasonable to add that measurements of cognitive potential, such as P300, may have their place in assessing auditory function and beyond. A patient’s practical function relies on more than simple auditory detection across a broad band of frequencies, including attention and cognition. For instance, high frequencies are the most susceptible to hearing loss [20,21] and are regularly implicated in those experiencing tinnitus [22]. Being able to employ physiological tests tailored to the specific hearing characteristics of a given patient may provide insight into their functional abilities.

P300 amplitude and latency do not exhibit consistent behaviors in response to the intensity and frequency characteristics of presented stimuli. Importantly, there is a dearth of research investigating P300 responses to frequencies beyond 4 kHz. In fact, it was shown when employing an oddball paradigm with 1500 Hz as standard and 1000 Hz as target/deviant stimuli, P300 amplitude did not significantly differ over the six intensities used (15, 25, 35, 45, 55, 65 dB SPL). On the other hand, P300 latencies showed a trend of a linear relation to stimulus intensities [23]. Another study employing two oddball paradigms, one with low frequencies (standard stimulus: 250 Hz; target stimulus: 500 Hz) and one with high frequencies (standard stimulus: 1000 Hz; target stimulus: 2000 Hz) presented at three intensities (45, 60, 75 dB SPL), found that increasing the stimulus intensity led to concomitant increases in P300 amplitude and decreases in P300 latency [24]. Interestingly, such dynamics were more pronounced for responses to standard stimuli, which may suggest that they were more tightly associated with auditory, but not cognitive, effects. The same authors also found that lower frequencies tended to produce longer latencies. Moreover, the N1 and P2 ERP components, in both oddball paradigms (low and high frequencies) showed larger amplitudes and shorter latencies with intensity increases, as well as some frequency effects [24]. In a later article, employing 1000 Hz as standard stimulus and 2000 Hz as a target stimulus, presented at 40, 60, 80 dB SPL, the authors reported that increases in stimulus intensity were linked to P300 latency decreases and amplitude increases [2]. Another study considered not only lower and a higher frequency P300 paradigms (standard stimulus: 250 Hz and target stimulus: 500 Hz; standard stimulus: 1000 Hz and target stimulus: 2000 Hz; presented at 40, 60 dB SPL), but also age effects in young vs. elderly individuals. This study reported that the younger sample exhibited larger amplitudes and shorter latencies than older participants [25]. Furthermore, the same authors found that P300 amplitudes were larger in response to low frequency compared to high frequency stimuli. In fact, P300 latencies were shorter in response to low frequency than high frequency stimuli, and as stimulus intensity increased, latencies decreased [25]. Finally, Cass and Polich [26] performed two different experiments with the aim of evaluating P300 changes in response to manipulations of stimulus intensity and frequency. In the first experiment, they reported that P300 amplitude increased and P300 latency decreased as the stimulus intensity increased. In the second experiment, which employed frequencies of up to 4000 Hz, they showed that P300 amplitude was affected only by the electrode site, but not the stimulus frequency.

Given the above uncertainty about the effects of intensity and frequency on P300 characteristics, the aim of the present study was to investigate P300 responses to the highest-frequency oddball paradigm reported to date in participants without hearing impairments. This study has the potential to identify more suitable testing conditions for hearing-impaired patients, who often present with high frequency hearing loss. Predictions: performing wave difference computations (response to deviant—response to standard stimuli) will prove to be a suitable method for disentangling sensory from cognitive responses. We expect that if no intensity effect is shown, P300 protocols can be employed in hearing-impaired patients (as long as the subject’s auditory thresholds are considered). Finally, it is important to highlight that this is the first study to consider P300 wave difference during an investigation of the response to high- and low frequency protocols and a further investigation of the stimulus intensity modulation of such responses.

## 2. Materials and Methods

### 2.1. Participants

The present preliminary study involved 11 participants without hearing impairments (8 female, 3 male; mean age = 23.909 ± 4.392 years). Hearing status was determined based on participants presenting with pure tone thresholds at the frequencies used in the study: 20 dB HL or lower. All participants provided informed consent and all study procedures were consistent with the Declaration of Helsinki and approved by the Institutional Review Board of the University of Colorado, Boulder.

### 2.2. Electroencephalographic (EEG) Recordings

#### 2.2.1. Stimuli

Participants were presented with two separate oddball paradigm conditions: high- and low frequency. Standard and deviant frequencies were chosen to exhibit half-octave differences in both conditions. The high frequency condition included a standard stimulus at 6000 Hz and a deviant stimulus at 8000 Hz, while the low frequency condition standard was 375 Hz and its deviant was 500 Hz. Deviant stimuli were presented at a rate of 20% probability. Standard and deviant stimuli were pseudorandomized over the course of the session within each condition, with standard and deviant stimuli being presented 200 and 50 times, respectively. Each high- and low frequency stimulus was 100 ms in duration, with 20 ms onset and offset ramps. All stimuli were calibrated to meet pitch perception curves. The interstimulus intervals for both conditions were 700 ms.

#### 2.2.2. Parameters and Procedures

During the research sessions, participants were seated in a comfortable chair in a sound-attenuating booth. Evoked potentials were collected using a Compumedics NeuroScan Synamps 2 system. Participants were fitted with several silver/silver chloride active electrodes, which were referenced to paired mastoids, as well as a two-electrode eye channel (lateral outer canthus—superior orbital), and a ground electrode (placed on the forehead). Active electrodes were placed at Fz, Cz, Pz, T3, and T4, according to the International 10–20 system. Electrode impedances were kept below 10 kOhms during all recordings.

Participants were instructed to pay attention to the deviant stimuli and to press a button when they heard a deviant sound. All experimental conditions were recorded with the eyes open [26]. Each paradigm was presented at four separate intensity levels: 20, 30, 40, and 50 dB HL [23,25]. Each intensity level was repeated at least twice in order to evaluate replicability. Participants were offered a 10 min break after the first 4 intensity trials.

### 2.3. EEG Analysis

During recording, EEG signals were bandpass-filtered between 0.01 and 30 Hz. Trials in which the EEG or EOG exceeded +100 uV were rejected. Then, the remaining sweeps associated with standard and deviant stimuli were averaged separately off-line. At least 200 standard and 50 deviant trials were retained and averaged following artifact rejection for each participant.

P1 was considered to be the most positive peak, occurring at between 50 and 90 ms [27]; N1 was considered the most negative peak, between 70 and 170 ms; and P2 was considered the most positive peak, between 140 and 270 ms post-stimulus onset; in case of a bifid or broad peak, the midpoint of the waveform was used [28]. All peak component amplitudes (P1, N1, P2) were measured from baseline to peak, or the midpoint of broad peaks. Latencies were chosen at the highest amplitude of the peak, or the midpoint of broad, flat peaks. Epoched data were baseline-corrected to the pre-stimulus interval of 100 ms and initial artifact rejection was performed at ±100 μV, according to a statistical threshold method for muscular, instrumental, and ocular artifacts rejection, which has been explored in many studies (e.g., [29,30,31]). EEG processing was performed by employing EEGlab [32] on MATLAB (Mathworks, R2012a) and SCAN (Compumedics Neuroscan, Inc., version 4.3) software. CAEP waveform peak components were visually identified and averaged after this step [27].

For the calculation of P300, previously established guidelines for elicitation and recording were taken into account [1]. Additionally, responses to standard and deviant tones were averaged independently. Then, the average of the response to deviant stimuli was subtracted from the response to standard stimuli for P300 [33]. P1, N1, P2, and P300 amplitudes and latencies were computed for both standard and deviant averaged waveforms, but for the former (P1, N1, P2), the responses were averaged, whilst for P300, a subtraction wave was performed using the responses to deviant and standard stimuli. Specifically, P300 latency was determined by marking the time at maximum positive waveform amplitude following the P1-N1-P2 complex in the deviant waveform (for absolute latencies) and in the difference waveform.

### 2.4. Statistical Analysis

Non-parametric statistics were applied, employing the software JASP (Version 0.19.1). In particular, Friedman’s ANOVA was employed to assess the influence of the variable “intensity” (4 levels: 20, 30, 40, 50) for each of the two oddball paradigms. Furthermore, Conover’s post hoc test was employed for post hoc comparisons on statistically significant effects; Holm’s correction was also employed.

## 3. Results

### 3.1. P1

#### 3.1.1. P1 Latency

We observed no statistical difference across intensities for the low frequency oddball paradigm (df = 3, Kendall’s w = 0.249, *p* = 0.081). On the other hand, in the high frequency paradigm, we found that intensity had the most significant effect (df = 3, Kendall’s w = 0.565, *p* = 0.008). Post hoc, pairwise analysis showed that 20 dB intensity elicited longer P1 latencies than 40 dB (*p* = 0.002) and 50 dB (*p* = 0.009) (Figure 1).

#### 3.1.2. P1 Amplitude

For P1 amplitude, intensity was not a significant factor in either the low- (df = 3, Kendall’s w = 0.216, *p* = 0.208) or high frequency paradigm (df = 3, Kendall’s w= 0.216, *p* = 0.208) (Figure 2).

### 3.2. N1

#### 3.2.1. N1 Latency

We found that intensity had the most statistically significant effect (df = 3, Kendall’s w = 0.506, *p* = 0.003) on the low frequency paradigm. Specifically, our analysis revealed that the 20 dB stimuli elicited longer latencies than those at 40 dB (*p* = 0.011) and 50 dB (*p* = 0.002). Additionally, the 30 dB stimuli elicited longer latencies than 40 dB (*p* = 0.038) and 50 dB (0.011) presentations. Conversely, for the high frequency paradigm, we did not observe any statistical differences across intensities (df = 3, Kendall’s w = 0.183, *p* = 0.223) (Figure 3).

#### 3.2.2. N1 Amplitude

N1 amplitude did not significantly differ by intensity for either the low frequency (df = 3, Kendall’s w = 0.012, *p* = 0.954) or high frequency (df = 3, Kendall’s w = 0.244, *p* = 0.119) paradigms (Figure 4).

### 3.3. P2

#### 3.3.1. P2 Latency

No statistical differences were observed by intensity for the low frequency paradigm (df = 3, Kendall’s w = 0.170, *p* = 0.204), though a statistically significant effect of intensity was seen for the high frequency paradigm (df = 3, Kendall’s w = 0.362, *p* = 0.034). More specifically, post hoc analysis showed that 20 dB intensity elicited longer P2 latencies than 30 dB (*p* = 0.026) (Figure 5).

#### 3.3.2. P2 Amplitude

In the low frequency paradigm, we observed that intensity had a statistically significant effect (df = 3, Kendall’s w = 0.516, *p* = 0.003). Looking more closely via post hoc analysis, we noted that the 50 dB intensity elicited higher P2 amplitudes than all the other intensities: 20 dB (*p* < 0.001), 30 dB (*p* = 0.003) and 50 dB (*p* = 0.048). Furthermore, we detected a statistically significant effect of intensity in the high frequency paradigm (df = 3, Kendall’s w = 0.456, *p* = 0.012). Again, post hoc analysis showed that the 50 dB intensity elicited higher P2 amplitudes than 20 dB (*p* = 0.005) (Figure 6).

### 3.4. P300

#### 3.4.1. P300 Latency

No statistical differences in latency were noted between intensity levels for either paradigm: low frequency (df = 3, Kendall’s w = 0.378, *p* = 0.079) or high frequency (df = 3, Kendall’s w = 0.150, *p* = 0.176; see Figure 7).

#### 3.4.2. P300 Amplitude

Similarly, peak amplitude did not differ by intensity for the P300—low frequency (df = 3, Kendall’s w = 0.278, *p* = 0.172); high frequency (df = 3, Kendall’s w = 0.048, *p* = 0.664) (Figure 8).

## 4. Discussion

The aim of the present study was to investigate P300 responses to high- vs. low frequency oddball paradigms in participants without hearing impairments in order to identify the most suitable testing conditions for hearing-impaired patients. We predicted that performing wave difference analysis for P300 would eliminate the sensory-only response while preserving more cognition-related components. Given the lack of significant findings obtained for P300 in response to the variation in intensity and the lack of influence of frequency, we posit that such a P300 protocol can be reliably employed in hearing-impaired patients. On the other hand, in obligatory CAEPs, we observed intensity- and frequency-related modulation, in accordance with previous evidence. These findings suggest an attentional bias in response to high frequency in earlier components, and separate attentional processes for later components.

More specifically, other researchers have previously demonstrated that the generators of the P1 CAEP component play a role in auditory stimulus detection and discrimination, as evidenced by associated systematic changes in P1 peak latency [34,35,36]. Additionally, in adults with mild hearing loss, it has been shown to contribute actively to compensatory mechanisms [27]. In the present study, however, the effects of intensity were only seen in the high frequency paradigm, even in a sample of young adults without hearing impairments. While speculatory at this point, one possible explanation for the observed intensity effect, which is present only in the high frequency condition, might be the discrimination and attentive reaction to these stimuli. That is, these stimuli could be evolutionarily perceived as potentially dangerous, which has been shown to be a characteristic typical of higher-frequency sounds [37,38]. In fact, for instance, higher frequencies are employed as warning cues in go/no go tasks [39], are the most commonly employed frequencies in fire-alarms [40], and are also used for conventional ambulance sirens and tonal backup alarms [37]. Moreover, in support of such hypothesis, in a motor task executed while participants were exposed to different tone frequencies, higher frequencies were rated by participants as more uncomfortable than lower ones, and such frequencies resulted in worsening in postural performance when participants were exposed to high frequencies in comparison to the quiet condition, a result that could possibly be explained by an alarm reaction triggered by the auditory stimuli [38]. Furthermore, the relationship between cortical activity in response to the exposure to pure-tone auditory stimuli, with the latter possibly being linked to ecological warning/alarm signals, has been suggested to be processed by the attentional network [41]. Thus, the earliest CAEP investigated in the present study, P1, consistent with previous evidence, is influenced by the exposure to threatening and emotional stimuli (attentional bias theory) at earlier stages of processing, while the modulation of later potentials, like P300, would bear a closer resemblance to conscious and evaluative processing of threat and emotion [42]. Furthermore, such topics are particularly relevant for eventual applications in hearing-impaired patients, given the evidence of altered EEG signatures of emotional cues detection in that population, both in children [30] and adults [43,44].

N1 is typically associated with a transition from mostly sensory to more cognitive processes and, in previous studies, has shown a decrease in latency for higher frequencies (4000 and 1000 Hz) compared to lower frequencies (250 Hz), as well as a tendency toward latency decreases with increasing stimulus intensities [45]. Similarly, in the present study, an effect of intensity was observed; specifically, a latency decrease for lower intensities (20 and 30 dB) compared to higher intensities (40 and 50 dB). Additionally, and notably, this investigation also included frequencies higher than those previously studied in an oddball paradigm, confirming that for these frequencies (higher than 4 KHz), there was no apparent effect of stimulation intensity. Previous evidence using various stimulus intensities and a frequency of 1000 Hz demonstrated similar intensity dependence for N1 and P2, with a linear trend showing higher intensities eliciting higher amplitudes [46]. The finding that P2 amplitude was influenced by stimulus intensity in both low- and high frequency oddball paradigms, alongside the analogous effects of intensity on N1 latency in the low frequency paradigm and P2 latency in the high frequency paradigm, aligns with the theory that N1 and P2 belong to the “vertex potential” (N1-P2) complex. This complex has been shown to be altered in tinnitus patients under different stimulus intensities, suggesting a lack of habituation to auditory stimulation [47].

In this context, the lack of significant N1 amplitude changes in response to the present paradigms, contrasted with the observed P2 amplitude changes, strongly supports the dissociation of N1 and P2 peaks under certain conditions, particularly in relation to attentional processes, as supported in recent research [5]. Moreover, some evidence indicated that as a participant’s attention level increases, N1 amplitude increases, while P2 amplitude decreases [5].

In previous reports, the P300 component has been shown to be affected by both stimulus intensity and cognitive load, such that lower intensities and higher cognitive load are associated with decreased P300 amplitudes and increased P300 latencies [48]. Other CAEP components show similar effects [49]. In general, our results suggest that earlier obligatory CAEP components were more affected by intensity than the P300 was. These findings have several implications. For instance, both decreased loudness and increased cognitive load are hallmarks of hearing loss, especially in adults [48]. According to our results, the early CAEPs would be more affected by stimulus intensity than later cognitive potentials in such individuals. By taking advantage of this distinction between early and late components, our results may imply that P300 paradigms could be used to separate loudness vs. the cognitive effects of hearing loss, consistent with [50,51]. That is, in conjunction with behavioral hearing testing, which often does not include assessment of cognitive aspects of hearing loss (especially in adults), characteristics of the P300 (e.g., latency) could be used to evaluate such effects, independent of more peripheral aspects of hearing loss. The present study shows that high frequency stimuli could be used in this type of paradigm, which is significant due to the commonality of high frequency hearing loss in adults.

## 5. Conclusions

To the best of our knowledge, this is the first article to report on the use of frequencies higher than 4 kHz (6 and 8 kHz) to elicit P300 responses. Our results support the notion that, given the lack of effect exerted by intensity on P300 parameters, such protocols can be employed in the testing of hearing-impaired patients, while taking into consideration the subject’s auditory threshold as a correction factor. Finally, the influence of stimulus intensity on CAEP in general was aligned with previous evidence showing that lower intensities elicited longer latencies and smaller amplitudes than higher intensities. One possible consideration is that higher frequencies could play a role in the response characteristics shown by N1 and P2, given their involvement in attentional processes linked to the detection of warning cues. Overall, we suggest that by altering stimulus factors (intensity and frequency), we can differentiate sensory/perceptual and more cognitive aspects of hearing, and that these techniques could potentially be used to provide more comprehensive clinical insight into patients’ hearing functionality. Of course, a limitation of the present study is represented by the limited sample size; therefore, an expanded version of the study involving a wider sample of participants would help to confirm the present results. Future studies should evaluate such claims in clinical populations, such as those with hearing loss and tinnitus.

## Figures and Tables

**Figure 1 brainsci-15-00209-f001:**
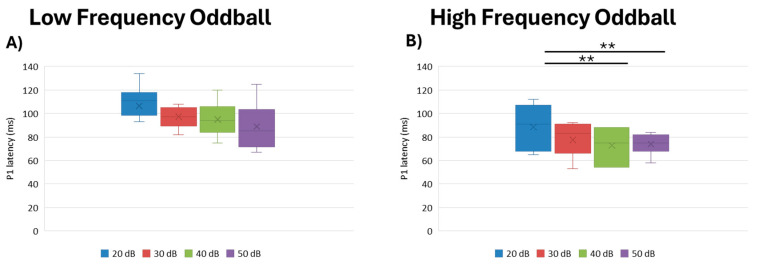
Boxplot representing the P1 latency in the two oddball paradigms: low frequency (**A**) and high frequency (**B**), in response to the different stimulus intensities employed in the study (20, 30, 40, 50 dB). In each box, “x” stands for the mean value and the line stands for the median value. ** stands for *p* ≤ 0.01.

**Figure 2 brainsci-15-00209-f002:**
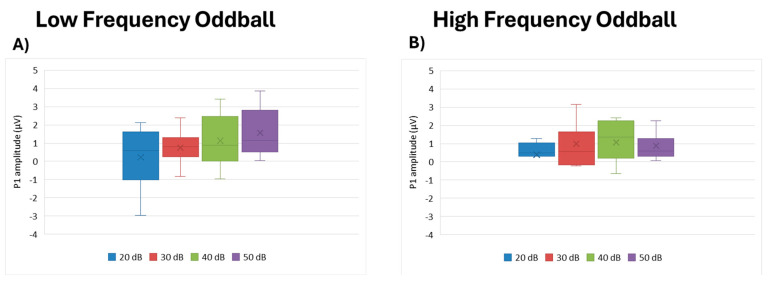
Boxplot representing the P1 amplitude in the two oddball paradigms: low frequency (**A**) and high frequency (**B**), in response to the different stimulus intensities employed in the study (20, 30, 40, 50 dB). In each box, “x” stands for mean value and the line stands for the median value.

**Figure 3 brainsci-15-00209-f003:**
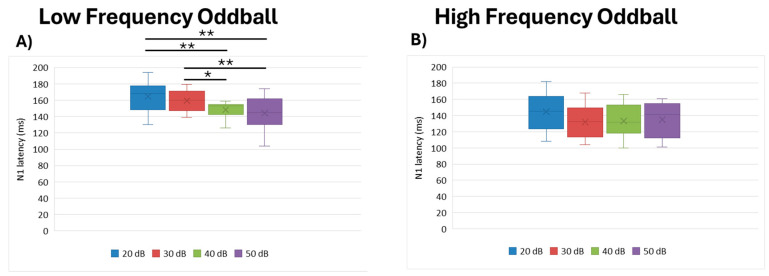
Boxplot representing the N1 latency in the two oddball paradigms: low frequency (**A**) and high frequency (**B**), in response to the different stimulus intensities employed in the study (20, 30, 40, 50 dB). In each box, “x” stands for the mean value and the line stands for the median value. ** stand for *p* ≤ 0.01; * stand for *p* ≤ 0.05.

**Figure 4 brainsci-15-00209-f004:**
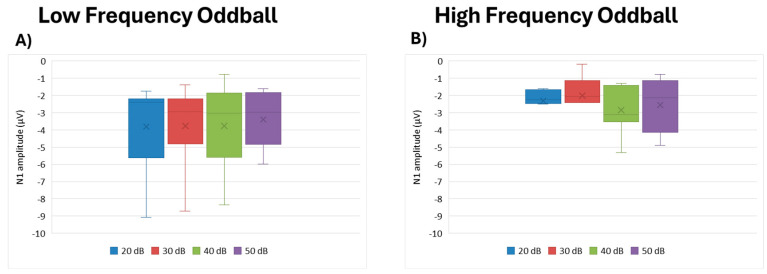
Boxplot representing the N1 amplitude in the two oddball paradigms: low frequency (**A**) and high frequency (**B**), in response to the different stimulus intensities employed in the study (20, 30, 40, 50 dB). In each box, “x” stands for the mean value and the line stands for the median value.

**Figure 5 brainsci-15-00209-f005:**
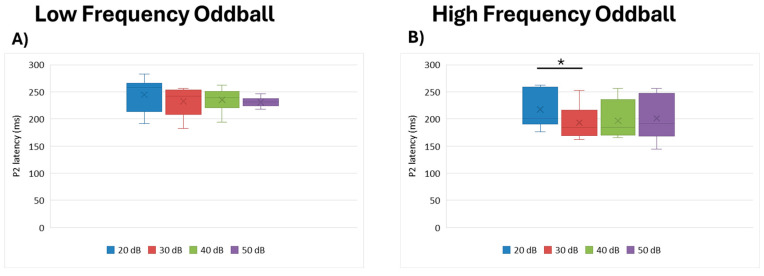
Boxplot representing the P2 latency in the two oddball paradigms: low frequency (**A**) and high frequency (**B**), in response to the different stimulus intensities employed in the study (20, 30, 40, 50 dB). In each box, “x” stands for the mean value and the line stands for the median value. * stands for *p* ≤ 0.05.

**Figure 6 brainsci-15-00209-f006:**
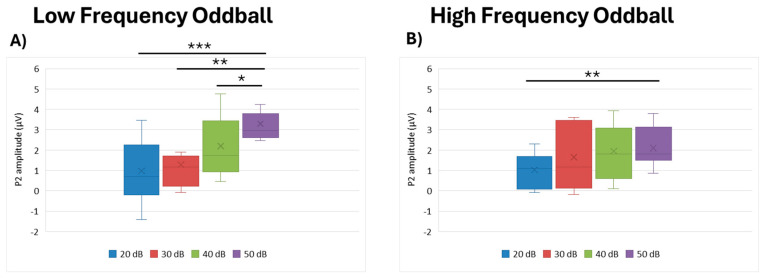
Boxplot representing the P2 amplitude in the two oddball paradigms: low frequency (**A**) and high frequency (**B**), in response to the different stimulus intensities employed in the study (20, 30, 40, and 50 dB). In each box, “x” stands for the mean value and line stands for the median value. *** stands for *p* ≤ 0.001; ** stands for *p* ≤ 0.01; * stands for *p* ≤ 0.05.

**Figure 7 brainsci-15-00209-f007:**
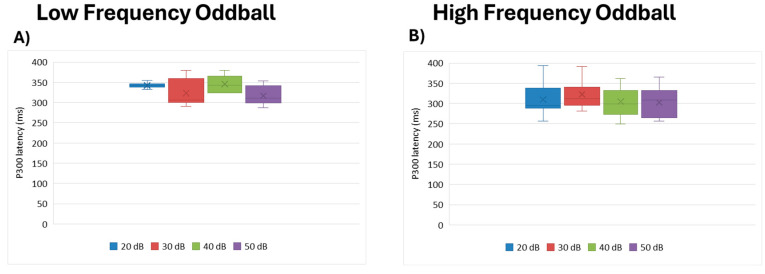
Boxplot representing the P300 latency in the two oddball paradigms: low frequency (**A**) and high frequency (**B**), in response to the different stimulus intensities employed in the study (20, 30, 40, and 50 dB). In each box, “x” stands for the mean value and the line stands for the median value.

**Figure 8 brainsci-15-00209-f008:**
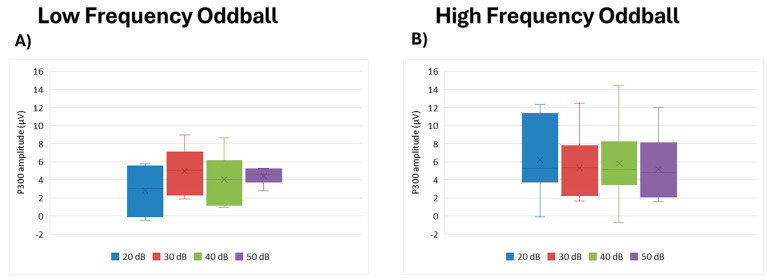
Boxplot representing the P300 amplitude in the two oddball paradigms: low frequency (**A**) and high frequency (**B**), in response to the different stimulus intensities employed in the study (20, 30, 40, and 50 dB). In each box, “x” stands for the mean value and the line stands for the median value.

## Data Availability

The data presented in this study are available on request from the corresponding author due to privacy restrictions.

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
