# Peer review of "P300 to Low and High Frequency Stimuli Are Not Influenced by Intensity in Adults with Normal Hearing"

_brainsci, 2025, doi:10.3390/brainsci15020209_

Round 1
Reviewer 1 Report
Comments and Suggestions for Authors
This manuscript investigates the effects of auditory stimulus intensity and frequency on P300 and cortical auditory evoked potentials (CAEP) to explore their potential applications in assessing hearing-impaired patients. The research question is meaningful, the experimental design is sound, and the data analysis is clear. However, there are weaknesses in the presentation and logical organization, and some conclusions lack sufficient support.
Major:
1、While the study highlights the scarcity of research on high-frequency stimuli, its findings align largely with existing trends (e.g., intensity effects on early CAEPs but not P300). There is a lack of groundbreaking discoveries. The hypothesis of “high-frequency stimuli as warning cues” is briefly mentioned but inadequately supported by the data or further discussion.
2、The sample size (11 participants, 3 males) is too small, and the gender imbalance raises concerns about the reliability and generalizability of the findings. The study lacks data from hearing-impaired individuals, limiting its clinical relevance.
3、The manuscript heavily relies on citing prior studies to support conclusions but lacks a deeper exploration of the mechanisms underlying the results. For instance, why are P300 characteristics unaffected by intensity and frequency? Is the hypothesis regarding high-frequency stimuli being “warning cues” supported by evolutionary or neurophysiological evidence?
4、The claim that P300 can separate sensory and cognitive effects is vague and lacks operational clarity.
5、Some sentences are overly complex and verbose, reducing readability.
6、Figures fail to illustrate key findings effectively. For example, while P300 results show no significant differences, trends or related data are not intuitively presented.
Comments on the Quality of English Language
Some sentences are overly complex and verbose, reducing readability.
Author Response
Comment 1:、While the study highlights the scarcity of research on high-frequency stimuli, its findings align largely with existing trends (e.g., intensity effects on early CAEPs but not P300). There is a lack of groundbreaking discoveries. The hypothesis of “high-frequency stimuli as warning cues” is briefly mentioned but inadequately supported by the data or further discussion.
Response 1: We thank the Reviewer for the comment and we agree about the alignment with existing trends, that supports the robustness of the present study. Concerning the lack of groundbreaking discoveries, we partially agree with the Reviewer, in fact, as reported at the beginning of the Conclusion section, “To the best of our knowledge this is the first article reporting on a study using frequencies higher than 4kHz (6 and 8 kHz) to elicit P300 responses”, therefore we believe that the study presents merit in terms of novelty.
Concerning the warning values of high frequency stimuli, we apologize for the perceived missing discussion underlined by the Reviewer. We expanded the concept in the Discussion section, despite of course the clear limitations of the study in terms of numerosity of high frequencies investigated and number of participants (please see the following point with respect to this). As stated in the Discussion section, such possible explanation is characterized by a speculatory nature, that we linked to some evidence borrowed from other studies. We added the following period in the Discussion section:
“In fact, for instance, higher frequencies are employed as warning cues in go/no go tasks [36], are the most actually employed frequencies in fire-alarms [37], and for conventional ambulance sirens and tonal backup alarms [34]. Moreover, in support to such hypothesis, in a motor task executed during the exposure to different tone frequencies, higher frequencies were rated by participants as more discomfortable than lower ones, and producing worsening in the postural performance when participants were exposed to high frequencies in comparison to the quiet condition, a result possibly explained by an alarm reaction triggered by the auditory stimuli [35]. Furthermore, the relationship between cortical activity in response to the exposure to pure tone auditory stimuli, the latter possibly linked to ecological warning/alarm signals, has been suggested to be processed by the attentional network [38].”
Comment 2:、The sample size (11 participants, 3 males) is too small, and the gender imbalance raises concerns about the reliability and generalizability of the findings. The study lacks data from hearing-impaired individuals, limiting its clinical relevance.
Response 2: We thank the Reviewer for the comment and we agree that the sample size is limited, in fact we employed non-parametric analyses and applied Holm correction to results. Such strategy should have mitigated the impact of the low numerosity on the type 1 error. Concerning the lack of data from hearing impaired individual, actually the study purposefully did not include such population, because the aim was instead to perform a preliminary analysis concerning the theoretical applicability of the present paradigm also on hearing impaired people, just correcting for their hearing threshold.
Concerning the gender imbalance, we thank the Reviewer for raising the point, in fact there are evidence of an influence of the gender on P300 amplitude, although it appears that P300 latency was comparable between genders [Melynyte, S., Wang, G. Y., & Griskova-Bulanova, I. (2018). Gender effects on auditory P300: A systematic review. International Journal of Psychophysiology, 133, 55-65.]. However, given the within group statistical analysis design, where the comparison among decibels was performed, the risk of the eventual influence of the gender was avoided. On the contrary, when the further studies comparing different groups will be performed, for sure we will need to take into account gender balance among groups. Thanks for highlighting such aspect!
Comment 3:、The manuscript heavily relies on citing prior studies to support conclusions but lacks a deeper exploration of the mechanisms underlying the results. For instance, why are P300 characteristics unaffected by intensity and frequency? Is the hypothesis regarding high-frequency stimuli being “warning cues” supported by evolutionary or neurophysiological evidence?
Response 3: We thank the Reviewer for the comment, please see the response to comment number 1 for the discussion of the present point.
Comment 4:、The claim that P300 can separate sensory and cognitive effects is vague and lacks operational clarity.
Response 4: We have clarified the operationality of the above claim and added to our claims regarding the novelty of the present paper (e.g., use of high frequency stimuli) toward the end of the Discussion by adding a reference in the last sentence and after that adding the following text:
“By taking advantage of this distinction between early and late components, our results may imply that P300 paradigms could be used to separate loudness vs. cognitive effects of hearing loss, consistent with [48] and [49]. That is, in conjunction with behavioral hearing testing, which often does not include assessment of cognitive aspects of hearing loss (esp. in adults), characteristics of the P300 (e.g., latency) could be used to evaluate such effects, independent of more peripheral aspects of hearing loss. The present study shows that high frequency stimuli could be used in this type of paradigm, which is significant due to the commonality of high frequency hearing loss in adults.”
Comment 5:、Some sentences are overly complex and verbose, reducing readability.
Response 5: We have re-worked many sentences throughout the paper to improve readability.
Comment 6:、Figures fail to illustrate key findings effectively. For example, while P300 results show no significant differences, trends or related data are not intuitively presented.
Response 6: We thank the Reviewer for the valuable comment. We turned all the figures in boxplot, so to provide a more descriptive representation of the data.
Reviewer 2 Report
Comments and Suggestions for Authors
The main objective of the presented manuscript was to show whether the loudness of auditory stimuli affects the P300 component reflecting cognitive processes. This is important for the use of the P300 component as a potential biological marker of cognitive processes in hearing impaired individuals. The novelty of the work is the use of a difference wave between the standard and the deviant, which according to the authors allows to separate sensory and cognitive components, as well as the use of stimuli of high frequency range, often disturbed in hearing impairment. Thus the materials presented in the manuscript are relevant and interesting in the light of the development of a system of biological markers of cognitive function.
Nevertheless, there are some small remarks and wishes for the work.
Title of the article is “P300 latency and amplitude are not influenced by stimulus intensity or frequency in adults with normal hearing.” I think the title should be changed. The title suggests a comparison of P300 responses at different frequencies, but the paper doesn't directly compare the P300 response at different frequencies. Perhaps the title ‘P300 to low and high frequency stimuli are not influenced by intensity in adults with normal hearing’ is more reflective of the content of the paper.
Line 27-28 “Obligatory CAEP components responded more readily to stimulus frequency and intensity changes, and one possible consideration is that higher frequencies could play a role in the response characteristics showed by N1 and P2, given their involvement in attentional processes linked to the detection of warning cues”. There seems to be an error in this phrase, because there were no differences in N1 amplitude and latency for high frequency.
In the introduction, I would recommend emphasising that in the studies reviewed (lines 79-106) of the P300 sensory and cognitive responses are mixed, as they looked at the amplitude of the components themselves on standards and deviants, rather than the differences between them. This will emphasise the relevance of your approach for assessing cognitive processes.
The paper uses stimuli intensity between 20 and 50 dB, which is lower than most previous work, I would like to see some justification for the choice of these frequencies.
Line 151 “Each intensity level was repeated at least twice in order to evaluate replicability. Participants were offered a break of 10-minutes after the first 4 intensity trials.” Do I understand correctly that there were totally 400 standard stimuli and 100 deviants for each volume and frequency? In the methodology, it would be desirable to specify how many trails were left in the end after removing artefacts for averaging.
Line 164 “Artifacts such as ocular and other extraneous muscle movement identified as separate components were removed from the data” - It is necessary to explain how these artefacts were identified as separate components and removed. If ICA or other analyses were performed, please specify. It is also necessary to specify in what software EEG and ERP processing was performed.
It is not clear from the methodology whether all both standard and deviant stimuli were averaged to analyse the P1, N1, P2 components; this point needs to be clarified.
Lines 168-171 “For the calculation of P300, guidelines for its elicitation and recording were taken into account [1], and the average of respectively latencies and amplitudes was performed for standard and deviant stimuli for each run. Responses to the standard and deviant tones were averaged. Then the average of the response to deviant stimuli were subtracted of the response to standard stimuli [29].” Some clarification needs to be made regarding the P300 analysis. Were responses to the standard and deviant tones averaged independently for each type of stimuli? It is necessary to clarify how the latency and amplitude of P300 in the difference wave were determined, the referenced paper [29] also does not describe this step.
It is recommended to specify a small sample in the limitations of the work. For example, in Figure 2 for low frequency you can see an increase in amplitude with increasing loudness, it is likely that with a larger sample these changes would become significant.
Author Response
Comment 1: Title of the article is “P300 latency and amplitude are not influenced by stimulus intensity or frequency in adults with normal hearing.” I think the title should be changed. The title suggests a comparison of P300 responses at different frequencies, but the paper doesn’t directly compare the P300 response at different frequencies. Perhaps the title ‘P300 to low and high frequency stimuli are not influenced by intensity in adults with normal hearing’ is more reflective of the content of the paper.
Response 1: We thank the Reviewer for the suggestion, we changed it accordingly
Comment 2: Line 27-28 “Obligatory CAEP components responded more readily to stimulus frequency and intensity changes, and one possible consideration is that higher frequencies could play a role in the response characteristics showed by N1 and P2, given their involvement in attentional processes linked to the detection of warning cues”. There seems to be an error in this phrase, because there were no differences in N1 amplitude and latency for high frequency.
Response 2: We thank the Reviewer for the comment and apologize for the lack of clarity. In fact, as the Reviewer outlined, N1 latency was modulated by the stimulus intensity only in the low frequency paradigm, while the N1 amplitude was not modulated in neither the protocols. Moreover, the P2 latency was modulated only in the high frequency protocol and finally P2 amplitude was modulated both in the low and high frequency protocols. Concerning N1 latency we meant that the lack of modulation in the high frequency protocol could be per se a response characteristic; whilst concerning N1 amplitude, we totally agree and corrected the abstract accordingly. Thanks for highlighting this.
Comment 3: In the introduction, I would recommend emphasising that in the studies reviewed (lines 79-106) of the P300 sensory and cognitive responses are mixed, as they looked at the amplitude of the components themselves on standards and deviants, rather than the differences between them. This will emphasise the relevance of your approach for assessing cognitive processes.
Response 3: We really thank the Reviewer for the suggestion, we added a sentence specifying this point in the Introduction section as follows at the end of the Introduction section:
“Finally, it is important to highlight that this is the first study employing P300 wave difference during the investigation of the response to high and low frequency protocols and with the further investigation of the stimulus intensity modulation on such responses.”
Comment 4: The paper uses stimuli intensity between 20 and 50 dB, which is lower than most previous work, I would like to see some justification for the choice of these frequencies.
Response 4: We thank the Reviewer for the comment, our choice was made on the basis of previous works employing such sound intensities on normal hearing participants, we added the references in the text.
Comment 5: Line 151 “Each intensity level was repeated at least twice in order to evaluate replicability. Participants were offered a break of 10-minutes after the first 4 intensity trials.” Do I understand correctly that there were totally 400 standard stimuli and 100 deviants for each volume and frequency? In the methodology, it would be desirable to specify how many trails were left in the end after removing artefacts for averaging.
Response 5: Consistent with lab protocols and the fact that participants were healthy young adults, at least 50 deviant trials remained following artifact rejection, otherwise the participant’s trace would have been excluded from the analysis. This number is in keeping with many published reports. We have added text describing the above on line 156.
Comment 6: Line 164 “Artifacts such as ocular and other extraneous muscle movement identified as separate components were removed from the data” - It is necessary to explain how these artefacts were identified as separate components and removed. If ICA or other analyses were performed, please specify. It is also necessary to specify in what software EEG and ERP processing was performed.
Response 6: We apologize for the miscommunication. We have revised this section to reflect our artifact rejection procedures, which consisted of rejecting EEG data with peaks that exceeded + 100 μV. See line 174:
“Epoched data were baseline corrected to the pre-stimulus interval of 100 ms and initial artifact rejection performed at ±100 μV, according to a statistical threshold method for muscular, instrumental and ocular artifacts rejection extensively published (e.g. [27], [28], [29]). EEG processing was performed employing EEGlab [32] on MATLAB and Neuroscan Scan software.”
Comment 7: It is not clear from the methodology whether all both standard and deviant stimuli were averaged to analyse the P1, N1, P2 components; this point needs to be clarified.
Response 7: We apologize for the lack of clarity. We have added text between lines 190-194 in the 2.3 section to address the Reviewer’s comment:
“Additionally, responses to standard and deviant tones were averaged independently. Then the average of the response to deviant stimuli were subtracted of the response to standard stimuli for P300 [32]. P1, N1, P2, and P300 amplitudes and latencies were computed for both standard and deviant averaged waveforms, but for the formers (P1, N1, P2) the responses were averaged, whilst for P300 was performed the subtraction wave between the responses to deviant and standard stimuli.”
Comment 8: Lines 168-171 “For the calculation of P300, guidelines for its elicitation and recording were taken into account [1], and the average of respectively latencies and amplitudes was performed for standard and deviant stimuli for each run. Responses to the standard and deviant tones were averaged. Then the average of the response to deviant stimuli were subtracted of the response to standard stimuli [29].” Some clarification needs to be made regarding the P300 analysis. Were responses to the standard and deviant tones averaged independently for each type of stimuli? It is necessary to clarify how the latency and amplitude of P300 in the difference wave were determined, the referenced paper [29] also does not describe this step.
Response 8: As in the above response, we regret that the initial version was unclear. We have added text between lines 194-197 in the 2.3 section to address the Reviewer’s comment:
“P1, N1, P2, and P300 amplitudes and latencies were computed for both standard and deviant averaged waveforms. Specifically, P300 latency was determined by marking the time at maximum positive waveform amplitude following the P1-N1-P2 complex in the deviant waveform (for absolute latencies) and in the difference waveform”
Comment 9: It is recommended to specify a small sample in the limitations of the work. For example, in Figure 2 for low frequency you can see an increase in amplitude with increasing loudness, it is likely that with a larger sample these changes would become significant.
Response 9: We thank the Reviewer for the comment and we agree that the sample size is limited, in fact we employed non-parametric analyses and applied Holm correction to results. Such strategy should have mitigated the impact of the low numerosity on the type 1 error. We added a sentence concerning the limits of the study with respect to the sample size in the conclusion:
“Of course, a limit of the present study is represented by the limited sample size, therefore an enlargement of the study involving a wider sample would help in confirming the present results”
Round 2
Reviewer 1 Report
Comments and Suggestions for Authors
The author has answered my question, and I have no other doubts.